# Pyruvate Kinase M2: A New Biomarker for the Early Detection of Diabetes-Induced Nephropathy

**DOI:** 10.3390/ijms24032683

**Published:** 2023-01-31

**Authors:** Yeon Su Park, Joo Hee Han, Jae Hyeon Park, Ji Soo Choi, Seung Hyeon Kim, Hyung Sik Kim

**Affiliations:** 1School of Pharmacy, Sungkyunkwan University, Suwon 16419, Republic of Korea; 2St. Mark’s School, 25 Marlboro Rd, Southborough, MA 01772, USA

**Keywords:** diabetes, nephropathy, pyruvate kinase M2, biomarker, urinary excretion

## Abstract

Diabetic nephropathy (DN) is a common complication of diabetes. DN progresses to end-stage renal disease, which has a high mortality rate. Current research is focused on identifying non-invasive potential biomarkers in the early stage of DN. We previously indicated that pyruvate kinase M2 (PKM2) is excreted in the urine of rats after cisplatin-induced acute kidney injury (AKI). However, it has not been reported whether PKM2 can be used as a biomarker to diagnose DN. Therefore, we try to compare whether the protein PKM2 can be detected in the urine samples from diabetic patients as shown in the results of DN models. In this study, high-fat diet (HFD)-induced Zucker diabetic fatty (ZDF) rats were used for DN phenotyping. After 19 weeks of receiving a HFD, the DN model’s blood glucose, blood urea nitrogen, and serum creatinine levels were significantly increased; severe tubular and glomerular damages were also noted. The following protein-based biomarkers were increased in the urine of these models: kidney injury molecule-1 (KIM-1), neutrophil gelatinase-associated lipocalin (NGAL), and PKM2. PKM2 had the earliest detection rate. In the urine samples of patients, PKM2 protein was highly detected in the urine of diabetic patients but was not excreted in the urine of normal subjects. Therefore, PKM2 was selected as the new biomarker for the early diagnosis of DN. Our results reflect current knowledge on the role of PKM2 in DN.

## 1. Introduction

Diabetes mellitus (DM) is a serious and complex metabolic disorder. It is one of the greatest socioeconomic challenges worldwide owing to its high incidence rate [1,2]. The morbidity and mortality of DM have been increasing in the last 20 years. In addition, the global prevalence of diabetes is predicted to increase to 439 million adults by 2030 [3]. Severe complications of DM are associated with end-stage renal disease and the requirement of transplantation or renal replacement therapy [4,5]. A particularly serious complication of DM is diabetic nephropathy (DN), which develops in nearly one-third of patients with type 2 diabetes [6].

Proteinuria is the gold standard marker for diagnosing diabetes-induced kidney injury [7]. However, in approximately 30% of patients with diabetes, kidney dysfunction is dramatically reduced before proteinuria is detected, making it inadequate for monitoring the progression of DN [8,9]. Recently, various biomarkers of DN have been identified, including kidney injury molecule-1 (KIM-1), cystatin C, neutrophil gelatinase-associated lipocalin (NGAL), and inflammatory markers. Even so, serum creatinine (SCr) and blood urea nitrogen (BUN) are currently used as indicators of DN in clinical trials; however, these indicators have low sensitivities for the early detection of DN.

Blood concentrations of SCr and BUN become pathological only after severe renal histopathological injury [10,11]. Therefore, several studies have aimed to develop new biomarkers for detecting the early stages of DN [12,13,14]. For biomarkers of DN to be effective, they must be easily measured using biological samples, and their detection must be directly related to specific tissue damage in clinical trials. Therefore, more promising urinary biomarkers are needed to detect DN as an early intervention strategy for diabetic complications.

Recently, the glycolytic enzyme pyruvate kinase M2 (PKM2) has emerged as a key enzyme that connects glycolysis with inflammation during acute kidney injury (AKI) [15,16]. Among pyruvate kinases (L, R, M1, and M2), PKM2 is highly expressed in podocytes and plays a critical role in the pathogenesis of AKI [17,18]. Moreover, PKM2 is considered a therapeutic target in DN [19], and it may serve as a protective protein against diabetes-induced renal failure. PKM2 also contributes to diabetes-related microvascular diseases, regulatory roles, and post-translational modifications [20]. In most of these studies, PKM2 was not introduced as a biomarker for the early detection of DN in clinical practice. In previous studies, we identified PKM2 as an early biomarker of AKI. We also showed that urinary excretion of PKM2 has a better predictive value for cisplatin-induced AKI than KIM-1, and NGAL [21].

Additionally, increased PKM2 expression is closely associated with damage to renal tubular epithelial cells [22,23,24,25]. Therefore, PKM2 detection in urine after renal tubular injury results from increased intratubular pressure. In rat ureteral obstructive AKI models, urinary excretion of PKM2 was significantly increased in damaged renal tubular cells [26]. Even so, the clinical usefulness of PKM2 in DN is yet to be demonstrated. Therefore, we investigated the diagnostic value of PKM2 as a urinary biomarker for the early detection of DN. This is the first study to demonstrate the relationship between urinary PKM2 levels and DN pathogenesis.

## 2. Results

### 2.1. Body and Organ Weight Changes in Rats with HFD Induced Diabetes

During the experimental period, the body weight and food intake of all animals were measured weekly. From pre-administration to 12 weeks after administration, changes in the body weight of the Zucker diabetic fatty (ZDF) rats were significantly higher than those of normal rats (Figure 1A). In addition, blood glucose levels continuously increased after 6 weeks in the HFD group and were significantly higher than those in the ND group (Figure 1B). As food intake is related to changes in body weight, we evaluated food intake during the experimental period. Food intake in the HFD group was higher than that in the ND group (Figure 1C). The kidney and liver weights were significantly higher in the HFD group than in the ND group (Figure 1D). Our results are similar to previously reported data which showed that hyperglycemic conditions increase the weight of the liver and kidneys [27]. In contrast, pancreatic weight was significantly lower in the HFD group than in the ND group.

### 2.2. Histopathological Examination and Serum Biochemical Parameters

Histological scores for proximal tubular and glomerular damage were calculated on the basis of hematoxylin and eosin (H&E) staining results. In ND-fed rats, morphological changes in the glomeruli and proximal tubules did not cause any damage to the kidneys (Figure 2A). However, glomerular and proximal tubular injuries, such as glomerular hyaline droplets, increased mesangial matrix, basophilic renal tubules, hyaline casts, and tubular dilatation, increased in HFD-fed ZDF rats (Figure 2B). Serum biochemical analysis was performed to assess organ toxicity caused by HFD-induced diabetes progression. The levels of serum biomarkers of nephrotoxicity (BUN and sCr) were significantly higher in the HFD group than in the ND group (Figure 2C). Glycated hemoglobin (HbA1c) concentrations were measured to diagnose DM. Serum HbA1c levels were higher in the HFD group than in the ND group (Figure 2C). To evaluate the histopathological changes induced by diabetes more clearly, serum advanced glycation end product (AGE) levels were measured. As expected, serum levels of AGE were significantly higher in the HFD group than in the ND group (Figure 2C). In line with these findings, diabetes-induced kidney injury was evident in HFD-fed ZDF rats.

### 2.3. Urinary Biomarker Levels in HFD Rats

Urinary excretion of microalbumin plays a key role in the detection of diabetes-induced renal injury [28,29]. We observed that the urinary albumin-to-creatinine ratio was significantly higher in HFD-induced diabetic rats than in ND-fed rats (Figure 3A). Protein levels in the urine significantly increased with progressing hyperglycemic conditions, accompanied by significant decreases in urinary Cr, which indicated a significant alternation in the kidney function of HFD model rats (Figure 3A). These results indicate the presence of kidney injury in HFD-induced diabetic rats, as determined by histopathological evaluation. Even so, albumin, protein, and Cr levels in the urine were not significantly increased after 5 weeks in HFD-induced rats. To compare the sensitivity of these urinary biomarkers, we also measured protein-based biomarkers, including PKM2, KIM-1, and NGAL, in rat urine. As shown in Figure 3B, PKM2 expression was higher in HFD-induced diabetic rats than in ND-fed rats after 5 weeks. These results were confirmed using immunohistochemical (IHC) analysis, in which PKM2 and KIM-1 overexpression was observed after 5 weeks in HFD-induced diabetic rats.

### 2.4. Demographics and Clinical Characteristics

The clinical characteristics of the participants in this study are shown in Table 1.

Seventy patients were enrolled in this study. According to the inclusion criteria, the normal controls were relatively young, with a mean age of approximately 27.5 years (range, 20–35 years). In patients with diabetes, the mean age and duration of diabetes were 65.8 and 3 years, respectively, with BUN levels that were significantly more than normal values. Other clinical characteristics of the patients with diabetes were not detected, since patient information was obtained from an old sample. The reliability of KIM-1, NGAL, and PKM2 as biomarkers for DN was investigated in clinical trials. The urinary levels of KIM-1, NGAL, and PKM2 were noted to be markedly increased in patients with diabetes than in normal controls (Figure 4). These results suggest that KIM-1, NGAL, and PKM2 may be useful as noninvasive biomarkers for the early detection of DN

## 3. Discussion

DN is a serious and complex disease that leads to kidney failure worldwide. It accounts for nearly half of all new cases of chronic kidney disease (CKD) [30]. DN occurs in approximately 30–40% of patients with type 2 diabetes and in 15–25% of patients with type 1 diabetes [31]. By 2030, approximately 300 million people will experience diabetes-related complications [32]. With the increasing prevalence of DM, the current focus of research is the identification of novel biomarkers for the early detection of kidney disease in individuals with diabetes [33,34]. Microalbumin is classically considered a good biomarker for detecting diabetic kidney disease. However, microalbumin levels are also increased in other conditions, such as obesity, posture, exercise, diet, smoking, infection, and inflammation [35,36,37]. Therefore, albuminuria may not be suitable for the early detection of DN because other diseases can lead to microalbumin excretion in the urine. As we have demonstrated, although urinary albumin levels were not increased in HFD-induced diabetic rats after 5 weeks, the levels significantly increased after 12 weeks.

In this study, we investigated the possible use of the urinary protein PKM2 as a potent biomarker in the urine samples of diabetic patients. PKM2 is a key enzyme in aerobic glycolysis and is overexpressed in tumor tissues. The abnormally high levels of PKM2 promote the progression of the energy supply in tumor cells [38]. In addition, high PKM2 expression leads to increased glucose uptake and reduced oxygen consumption. This phenomenon is known as the Warburg effect in cancer cells [39]. Previously, we demonstrated that PKM2 is overexpressed in the damaged renal medulla and tubular epithelial cells in cisplatin-induced AKI rat models and streptozotocin-induced diabetic models [13,21]. Although efforts were made to characterize the role of PKM2 in renal function, particularly under hyperglycemia-induced renal injury [40,41], its contribution to DN pathophysiology remains largely unknown. Thus, this study further elucidates the importance of using PKM2 biomarkers for the early diagnosis of DN in patients with diabetes. In addition, the measurement of PKM2 in the urine is distinct from that in the plasma. Blood biomarkers necessitate venipuncture, may be influenced by systemic processes, and are often highly correlated with the glomerular filtration rate [42,43]. By contrast, urine collection is easy and painless, highlighting the utility of urine assays in clinical and research applications.

Furthermore, urinary excretion of PKM2 is likely able to predict DN progression. Thus, we compared the urinary excretion levels of PKM2 with routinely tested SCr and BUN levels in HFD-induced diabetic rats. To our knowledge, this study is the first to compare PKM2, SCr, and BUN as urinary biomarkers in diabetic rat models and human samples to identify a new approach for the detection of DN. However, we had limited sample numbers, so our age-adjusted and gender-adjusted comparisons did not match in the present study. If a large number of clinical samples are secured in a further study, it is expected that the correlations study between the urinary PKM2 level and the severity of DN in diabetic patients will be very important.

In the HFD-induced animal model, DN was diagnosed 12 weeks after the administration of HFD using BUN and SCr, according to a previous study [44]. Our data revealed that these renal injury biomarkers (BUN and SCr) were not increased in the serum of HFD-fed rats collected 5 weeks before a diagnosis of DN. However, PKM2 levels were significantly increased in the urine of rats 5 weeks after HFD administration. Furthermore, we compared the urinary excretion levels of PKM2 in patients with diabetes and those in healthy controls. Interestingly, the urinary excretion of PKM2 was also significantly elevated in patients with diabetes but not in normal controls. These findings are congruent with a previous study that demonstrated that the expression of PKM2 in podocytes of patients with diabetic kidney disease (DKD) is significantly higher than that in patients with other kidney diseases [45]. Another group reported that the expression of PKM2 in podocytes likely plays an important role in the pathogenesis of DKD [46].

Recent evidence suggests that KIM-1 may not only serve as a marker of injury but may also contribute to its pathology by promoting proximal tubule cell damage by facilitating fatty acid uptake [29]. Several studies have used KIM-1 and NGAL as sensitive biomarkers for the early detection of AKI [47,48,49]. A previous study of ours indicated that urinary PKM2 is a biomarker for the prediction of cisplatin-induced AKI [21]. In this study, we found that the time course of urinary PKM2 differed from those of KIM-1 and NGAL in models of HFD-induced diabetes. Moreover, PKM2 is not expressed in normal kidney tissues in rodents [50]. Therefore, we speculate that PKM2 leaked into the urine due to damaged renal tissues, at least in HFD-induced diabetic animal models. As shown in Figure 3, the urinary excretion of PKM2 after the HFD gradually increased. This suggests that renal PKM2 began leaking into the urine at 5 weeks and continued to leak until at least 19 weeks. In addition, the DN detection values of PKM2 were superior to those of KIM-1 and NGAL. Therefore, urinary PKM2 is useful in the detection and monitoring of DN.

The kidneys are not typically considered target organs of insulin action. Even so, PKM2 is predominantly expressed in tubular cells and, to a lesser extent, in mesangial and endothelial cells in the glomerulus [22,23]. In addition, PKM2 was detected early in renal tubular and interstitial injury during CKD progression [51]. However, although the present study showed that enhanced urinary excretion of PKM2 may be associated with DN progression, the involvement of PKM2 in DN is still uncertain. In rats with streptozotocin-induced diabetes, we found that urinary excretion of PKM2 was significantly increased, suggesting that PKM2 plays a detrimental role in DN [52].

Furthermore, DN progression is closely associated with oxidative stress and inflammation. Ahmad et al. [53] reported that prominent upregulation of proinflammatory cytokines might lead to pancreatic beta cell destruction and hyperglycemia in DM. In particular, the inflammation of glomerular endothelial cells promotes macrophage activation and leads to the development of DN. Thus, PKM2 may induce renal inflammation by promoting NF-kB activation in type 2 DN [54]. Therefore, further studies are required to elucidate the precise role of PKM2 as a target for detecting DN and to determine the urinary level of PKM2 in patients with diabetes so as to facilitate early intervention in DN progression.

## 4. Materials and Methods

### 4.1. Animal Experiments

ZDF rats (4 weeks old) were purchased from Woojung Scientific Animal Inc. (Seoul, Korea). All animals were housed with automatically controlled temperature (23 ± 2 °C) and a 12 h light/dark cycle in filtered-air laminar-flow cabinets. All procedures were approved by the Institutional Animal Care Committee of Sungkyunkwan University (approval number: SKKUIACUC2019-03-25-1; September 2020). Rats were randomly divided into two groups: (1) ND (*n* = 6) and (2) HFD (*n* = 18). The HFD group was further divided into three groups: (1) 5-week HFD group (*n* = 6), (2) 12-week HFD group (*n* = 6), and (3) 19-week HFD group (*n* = 6). The ND group was fed a cholesterol-free diet. The HFD, comprising proteins, fats, carbohydrates, fibers, minerals, and vitamins with 60 kcal% fat, was purchased from Research Diets, Inc. (New Brunswick, NJ, USA). During the experiment, blood glucose concentrations were measured weekly using a glucometer (ACCU-CHEK Performa). In addition, body weight was measured weekly during the experiments. Urine was collected from each animal kept in the metabolic cage overnight, and 24 h urine samples were collected at 5, 12, and 19 weeks from individual animals in containers (0.1% sodium azide). After sacrificing the rats, their major organs (liver, kidney, pancreas, and testis) were perfused with normal saline to remove residual blood and were stored at −80 °C (Figure 5).

### 4.2. Analysis of Biochemical Parameters

Blood was collected from the abdominal aorta and centrifuged at 3000× *g* for 10 min to collect serum. The sera were immediately stored at −80 °C to analyze biochemical parameters, including BUN and SCr, using an Olympus AU400 chemistry analyzer (Tokyo, Japan).

### 4.3. Analysis of Blood Glucose Levels

Fasting blood glucose levels were monitored weekly during the experimental period. Blood was collected from the tail vein in the morning, and blood glucose levels were estimated using a glucometer (ACCU-CHEK; Daeil Pharm. Co, Ltd., Seoul, Korea).

### 4.4. Measurement of HbA1c

Serum HbA1c levels were measured using commercial test kits (CUSABIO Biotech Co., Cat. #CSB-E08140r, Japan) according to the manufacturer’s instructions.

### 4.5. Measurement of AGE

According to the manufacturer’s instructions, AGE levels were determined using an enzyme-linked immunosorbent assay kit (Cell Biolabs, San Diego, CA, USA). The microplate was coated with a specific monoclonal antibody against AGEs and incubated with the standards and samples. The unbound conjugate was washed, and the plate was incubated for 1 h at 37 °C with *horseradish peroxidase (*HRP)-conjugated secondary antibodies. The substrate solution was added for 2–20 min, the color change was monitored, and the microplate was read at 450 nm.

### 4.6. Analysis of Urinary Parameters

Rats were maintained in individual metabolic cages for 24 h (under fasting conditions with free access to water). Subsequently, urine was collected under cool conditions and immediately centrifuged at 900× *g* and 4 °C for 10 min to remove insoluble materials and cellular debris. Next, the urine supernatant was aliquoted into sterile tubes and stored at –80 °C. Urinary abumin, Cr, and protein concentrations were analyzed using a Hitachi 7180 autoanalyzer (Hitachi, Tokyo, Japan).

### 4.7. Western Blot Analysis

The levels of various urinary protein-based markers, such as KIM-1 (1:1000), NGAL (1:1000), and PKM2 (1:1000), were analyzed using western blotting. Antibody against KIM-1 (PA5-79345) was purchased from Invitrogen (Waltham, MA, USA). NGAL (ab23477) and PKM2 (ab85555) antibodies were purchased from Abcam (Cambridge, Massachusetts, USA), while anti-mouse and anti-rabbit IgG, HRP-linked-conjugated secondary antibodies were procured from Santa Cruz Biotech. (Santa Cruz, California, USA). Briefly, the urine was centrifuged (1000× *g* for 10 min), and the supernatant (with the total urinary proteins) was collected and diluted in a 1:2 ratio with double-distilled water. Protein concentrations in the supernatants were estimated using the BCA protein assay (Thermo Fisher Scientific, Waltham, MA, USA). SDS/polyacrylamide was used to separate protein samples (60–80 µg). First, polyvinylidene fluoride (PVDF) membranes were blocked with a 5% milk buffer at 25 °C (room temperature) and incubated overnight with the primary antibodies at 4 °C. Next, PVDF membranes (Millipore, Burlington, MA, USA) were washed with Tris-buffered saline with 0.1% Tween-20 detergent and then incubated with secondary antibodies (1:10,000) at room temperature for 1 h. Protein expression and band intensity were measured using an enhanced chemiluminescence reagent kit (Amersham Biosciences Corp., Little Chalfont, UK).

### 4.8. Histological Examination

Kidney tissues were preserved immediately after collection in 10% neutral buffered formalin. Paraffin sections of these tissues (4 µm) were prepared for H&E staining to determine morphological abnormalities. At least five random positive areas in each section were imaged at 200× magnification.

### 4.9. IHC Analysis

IHC analysis was performed to investigate PKM2 expression. The slides were placed in a xylene chamber and hydrated in graded alcohol and water. The slices were moved to a chamber with 3% H_2_O_2_ to quench peroxidase activity. The samples were treated with 4% bovine serum albumin (BSA) for 1 h at 37 °C and were subsequently washed three times with Tris-buffered saline (TBS), then incubated with primary antibodies (PKM2, 1:250 and KIM-1, 1:300) at 4 °C overnight to prevent non-specific binding. After incubation with specific antibodies, the membranes were washed three times with TBS, a secondary antibody conjugated with HRP was added, and the sections were incubated at room temperature for 45 min. The slides were counterstained with hematoxylin after immunostaining with diaminobenzidine tetrahydrochloride as a visualizing agent.

### 4.10. Analysis of Clinical Urine Samples from Patients with Diabetes

Urine samples from 20 normal controls and 70 patients with diabetes were collected at Soonchunhyang University Hospital (Cheonan, Korea) between June 2016 and July 2016 to detect the candidate biomarkers (KIM-1, NGAL, and PKM2). Baseline clean-catch spot urine samples were obtained in sterile containers in the morning following overnight fasting according to standard methods and subsequently processed into 1.8 mL aliquots before storage. While preparing for analyses, samples were thawed and centrifuged at 1000× *g* for 20 min at 4 °C. The supernatant was used to perform a biomarker assay. In addition, the urine samples were subjected to western blot analysis. The patients provided written informed consent, and the collection of human urine samples was approved by the Medical Ethics Committee of Soonchunhyang University, School of Medicine and Hospital, Cheonan Korea (SCHCA 2015-12-023).

### 4.11. Statistical Methods

The experimental data were analyzed at least three times, and the results are expressed as the mean ± standard deviation. To determine statistical significance, we used a one-way analysis of variance and subsequently compared the results using Bonferroni’s multiple comparison tests (* *p* < 0.05, ** *p* < 0.01, indicating significant differences between the control and treatment groups). For non-normally distributed quantitative data, a comparison between the study and control groups was conducted using the Mann–Whitney test (nonparametric *t*-test). All statistical comparisons were performed using SigmaPlot graphing software and Statistical Package for Social Sciences version 13 (SPSS, Inc., Chicago, IL, USA).

## 5. Conclusions

In summary, our study demonstrated that the urinary excretion of PKM2 was significantly increased in HFD-induced diabetic experimental animals and patients with clinical DN. The measurement of PKM2 in the urine may be used as a novel biomarker for the early detection of diabetes-induced nephropathy. In the future, large-scale in-depth examinations of the viability of urinary protein PKM2 as a prognostic biomarker of DN are warranted to confirm the hypothesis of this study.

## Figures and Tables

**Figure 1 ijms-24-02683-f001:**
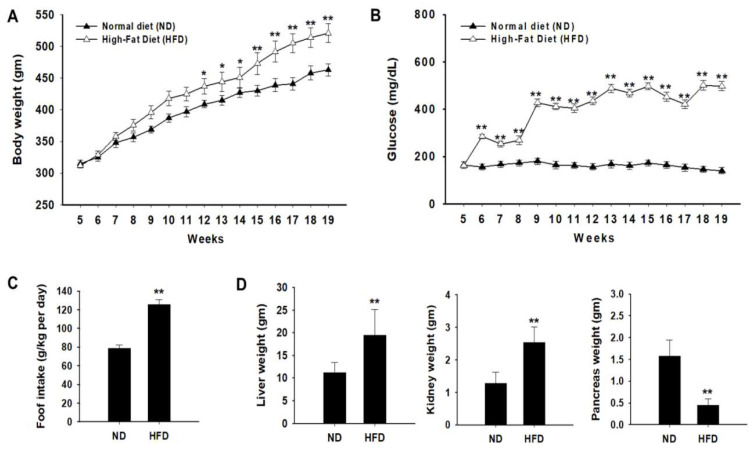
Effects of a high-fat diet (HFD) on body weight and organ weight. Zucker diabetic fatty (ZDF) rats were divided into two groups: (1) the normal diet (ND) and (2) HFD groups. (**A**) Weekly body weight changes were measured in each group. (**B**) Effect of HFD on fasting blood glucose concentration (mg/dL) was monitored. (**C**) Food intake was measured during the experimental period. (**D**) The liver, kidney, and pancreas weights were measured in each group. Data of each group are expressed as the mean ± standard deviation (SD). * *p* < 0.05; ** *p* < 0.01. ND, normal diet; HFD, high-fat diet.

**Figure 2 ijms-24-02683-f002:**
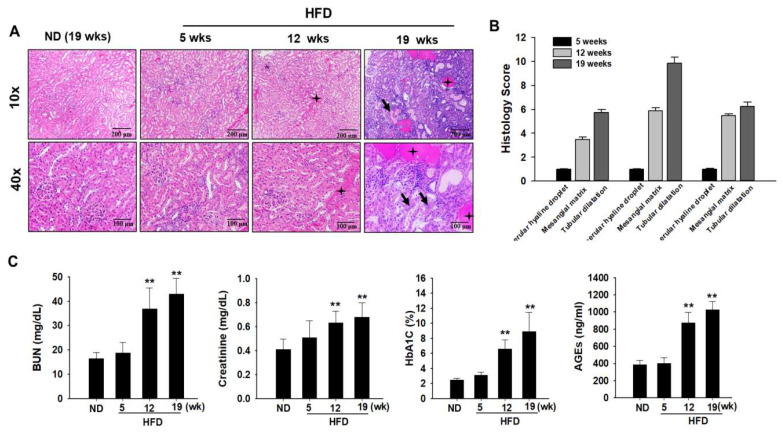
Effects of a high-fat diet (HFD) on biochemical parameter changes. Zucker diabetic fatty (ZDF) rats were divided into two groups: (1) the normal diet (ND) and (2) HFD groups. (**A**) Kidney samples were histologically examined. Renal histological abnormalities, including a hyaline cast of renal tubules, mesangial cell expansion, thickening of the basement membrane, increased mesangial matrix, formation of glomerular hyaline droplets (black stars), and tubular dilatation (black arrow), were observed in the kidney of diabetes-induced rats using hematoxylin and eosin (H&E) staining. (**B**) Histology scores for renal injury are as follows: 1, normal; 2, minimal; 3, slight; 4, moderate; 5, marked; and 6, severe at 5 weeks, 12 weeks, and 19 weeks in the kidneys of HFD-induced rats with DN. (**C**) Levels of blood urea nitrogen (BUN), serum creatinine (SCr), glycated hemoglobin (HbA1C), and advanced glycation end products (AGEs) in the blood serum were measured after fasting at the end of experiment. All abovementioned biochemical parameters were analyzed under 12 h fasting conditions. Data are expressed as the mean ± standard deviation of each group. ** *p* < 0.01. ND, normal diet; HFD, high-fat diet.

**Figure 3 ijms-24-02683-f003:**
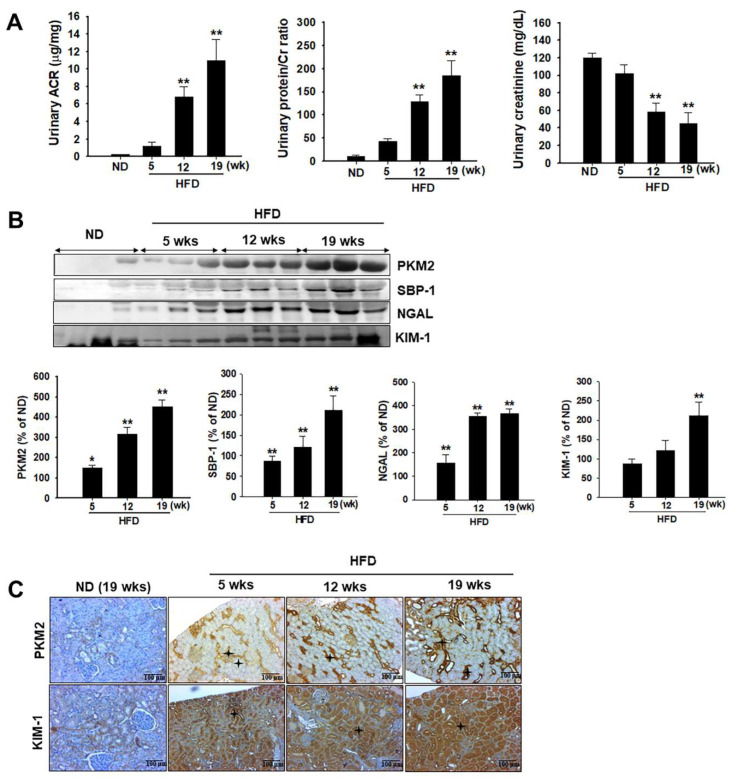
Effects of a high-fat diet (HFD) on urinary biochemical parameter changes. Zucker diabetic fatty (ZDF) rats were divided into two groups: (1) normal diet (ND) and (2) HFD groups. (**A**) Urinary albumin to creatinine (Cr) ratio, urinary protein to Cr ratio, and urinary Cr level were estimated at the indicated time points. (**B**) Urinary excretion of PKM2, SBP-1, NGAL, and KIM-1 levels were measured using western blot analysis. The semiquantitative levels of these biomarkers are indicated. Data are expressed as the mean ± standard deviation of each group. ** *p* < 0.01. (**C**) Representative immunohistochemical analysis of PKM2 and KIM-1 in the kidney of HFD-induced diabetic rats and normal diet-fed rats. * Black stars indicate PKM2 and KIM-1 expression in the damaged tubular tissues. ND, normal diet; HFD, high-fat diet.

**Figure 4 ijms-24-02683-f004:**
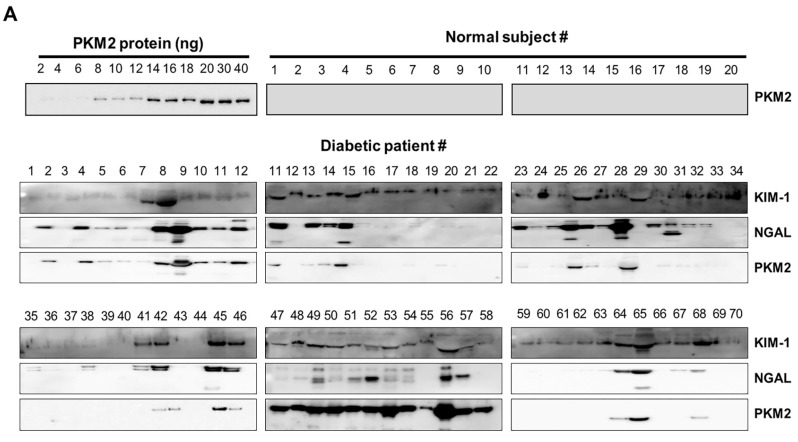
Urinary excretion of protein-based nephrotoxicity biomarkers in both normal subjects and patients with diabetes. (**A**) KIM-1, NGAL, and PKM2 levels were measured in the urine of healthy controls (*n* = 20) and patients with diabetes (𝑛 = 70) using western blot analysis. Urine proteins were resolved in 10% sodium dodecyl sulphate-polyacrylamide gel electrophoresis (SDS-PAGE) and analyzed using western blotting. (**B**) Densitometry analysis of the PKM2 band intensities in the urine of patients was normalized for 10 ng of recombinant protein PKM2.

**Figure 5 ijms-24-02683-f005:**
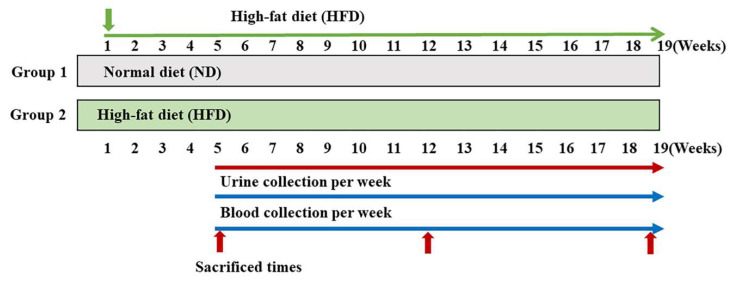
Experimental design. After 1 week of acclimatization, Zucker diabetic fatty (ZDF) rats were randomly split into two groups: one group received a standard chow diet (ND group, *n* = 6) and another received a high-fat diet (HFD group, *n* = 18). ND, normal diet; HFD, high-fat diet.

**Table 1 ijms-24-02683-t001:** Demographic and clinical characteristics of participants.

Characteristics	Normal Controls	Patients with Diabetes
Age (yr)	27.5 ± 3.87	65.8 ± 9.24
Sex		
Male	8	48
Female	12	22
Duration of diabetes (yr)		>3
Serum creatinine (mg/dL)	NA	NA
BUN (mg/dL)	NA	68.23 ± 23.57
BMI	22.3 ± 0.54	24.7 ± 3.24

BUN, blood urea nitrogen; BMI, body mass index; NA, not available.

## Data Availability

Not applicable.

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
