# Peer review of "Pyruvate Kinase M2: A New Biomarker for the Early Detection of Diabetes-Induced Nephropathy"

_ijms, 2023, doi:10.3390/ijms24032683_

Round 1

Reviewer 1 Report

Authors reported that PKM2 is a novel biomarker for Diabetic nephropathy from the rats and young adult. The work is well written and all aspects are clearly defined. This work will be very beneficial for the people who work in the area. I consider this work can be published in this form. However, please check the number of subtitle in section 4 materials and method, they are suppose to be 4.2 to 4.8

Author Response

Comments 1: Authors reported that PKM2 is a novel biomarker for Diabetic nephropathy from the rats and young adult. The work is well written, and all aspects are clearly defined. This work will be very beneficial for the people who work in the area. I consider this work can be published in this form. However, please check the number of subtitles in section 4 materials and method, they are supposed to be 4.2 to 4.8

Answer: Thank you for your kind comments. We corrected the number of subtitles in section 4 materials and methods.

Reviewer 2 Report

The Manuscript entitled Pyruvate Kinase M2: A New Sensitive Biomarker for the Early 2 Detection of Diabetic-Induced Nephropathy”. This is an intriguing research that the authors examined. They have demonstrated that the urinary excretion of PKM2 levels significantly increased in HFD-induced diabetic experimental animals and patients with clinical diabetic nephropathy. The study is well executed, but I have some minor concerns.

1.     The main concern is the similarity percentage, which is more than 50%; you should strive to minimize this up to 15%.

2.     To make a strong background for the study, could you add some latest studies on the function of PKM2 in nephropathy to the introductory section?

3.     In the section on material techniques, all biochemical tests are gathered under a single category; nevertheless, it would be more appropriate to separate them into their own sections.

4.     In the discussion portion of their papers, the authors need to include more research investigations suitably to fill in the gaps and make it easier to grasp for comprehension. 

5.     Grammatical and punctuation mistakes need to be corrected. Perform a careful and comprehensive check on the Manuscript.

6.     There are several abbreviations throughout the document. Please keep track of the abbreviations used in the text by using a list of abbreviations.

7.     Add the following important reference suitably in the discussion section.

https://pubmed.ncbi.nlm.nih.gov/30532634/

Author Response

Comments 1:     The main concern is the similarity percentage, which is more than 50%; you should strive to minimize this up to 15%.

Answer: Thank you for your comments, we revised the similarity of our text less than 10%.

Comments 2:     To make a strong background for the study, could you add some latest studies on the function of PKM2 in nephropathy to the introductory section?

Answer: Thank you for your comment, we added the function of PKM2 in nephropathy in the Introduction part.

Comments 3:     In the section on material techniques, all biochemical tests are gathered under a single category; nevertheless, it would be more appropriate to separate them into their own sections.

Answer: Thank you for your valuable comments. We separate the biochemical tests in Materials and methods part.

4.2. Analysis of Biochemical Parameters

Blood was collected from the abdominal aorta and centrifuged at 3000 × g for 10 min to collect serum. The sera were immediately stored at −80°C to analyze biochemical parameters, including BUN and SCr, using an Olympus AU400 chemistry analyzer (Tokyo, Japan).

4.3. Analysis of Blood Glucose Levels

Fasting blood glucose levels were monitored weekly during the experimental period. Blood was collected from the tail vein in the morning, and blood glucose levels were estimated using a glucometer (ACCU-CHEK; Daeil Pharm. Co, Ltd., Seoul, Korea).

4.4 Measurement of HbA1c

Serum HbA1c levels were measured using commercial test kits (CUSABIO Biotech Co., Cat. #CSB-E08140r, Japan) according to the manufacturer’s instructions.

4.5 Measurement of AGE

According to the manufacturer's instructions, AGE levels were determined using an enzyme-linked immunosorbent assay kit (Cell Biolabs, San Diego, CA, USA). The microplate was coated with a specific monoclonal antibody against AGEs and incubated with the standards and samples. The unbound conjugate was washed, and the plate was incubated for 1 h at 37°C with horseradish peroxidase (HRP)-conjugated secondary antibodies. The substrate solution was added for 2–20 min, the color change was monitored, and the microplate was read at 450 nm.  

Comments 4:     In the discussion portion of their papers, the authors need to include more research investigations suitably to fill in the gaps and make it easier to grasp for comprehension. 

Answer: Thank you for your valuable comments. We revised discussion parts as your comments.

Comments 5:     Grammatical and punctuation mistakes need to be corrected. Perform a careful and comprehensive check on the Manuscript.

Answer: Our paper clearly revised English correction by native speakers

Comments 6:     There are several abbreviations throughout the document. Please keep track of the abbreviations used in the text by using a list of abbreviations.

Answer: We added the abbreviations used in the text by using a list of abbreviations.

Comments 7:     Add the following important reference suitably in the discussion section.

https://pubmed.ncbi.nlm.nih.gov/30532634/

Answer: we added this reference in the discussion section. Zingerone (4-(4-hydroxy-3-methylphenyl) butan-2-one) protects against alloxan-induced diabetes via alleviation of oxidative stress and inflammation: Probable role of NF-kB activation. Saudi Pharm J. 2018 Dec;26(8):1137-1145.

Reviewer 3 Report

Dear Authors,

This paper concerned problem of early detection of diabetic nephropathy. This is interesting and important problem. But I have many commentaries and questions:

1.       Part 2.1- what for was this part – it is known that obese animals have higher weight and higher organ weight, what is new in this part?

2.       Lines 97-99- this sentence is not proper in English and difficult to understand; if there is no any damages- there is no morphological changes, there are two time ‘rats’ in one sentence.

3.       Line 102- the last part of the sentence is not logical- I think You mean ‘…identified in HFD-fed ZDF rats’ ?

4.       Figure 2 part B- it isn’t understandable. Why on the figure are three parts, what is mean?

5.       Table 1- this groups are not comparable, control is too small for statistic, the is lack of clinical characteristic, what is not proper for conclusions. There is mistake “BUM”, as I think? How was the DN diagnosed in this group, and how exclude other reasons of kidney damages.

6.       Line 143- according to inclusion criteria- where they are written? If in control group are almost only men, it is normal that will be higher mean BUN,

7.       Lines 145-146- it couldn’t be compared if we don’t have any information about subjects enrolled to control group.

8.       Lines 148-150- Did You exclude from the study patients with different other kidney diseases, what are the exclusion criteria for the study. If we want to find the early biomarker, we should have series of tests in the same group of patients and then compare what were different in group which later develop DN. All mention markers are controversial, and are not approved, despite for example NGAL is known for many years.

9.       Figure 4- If Authors compare to BUN- that marker seems not be very early but rather late.

10.   Lines 200—203- I couldn’t find information that Authors took the samples at 9 weeks.

11.   Lines 205-206- I couldn’t find information about taking samples at 9 weeks, and their results.

12.   Lines 224-226- there is no bibliography to this sentence

13.   Part 4.1- there is lack of diabetes recognition criterion in rats

14.   Lines 238-239- There is no information how the blood samples were taken

15.   Lines 239-241- there is no information at what time point were urine samples collected

16.   Line 300- How many control subjects You enroll to the study (in the text are 20, in the table 8 person. What are inclusion and exclusion criteria. These are not comparable groups, and lack of subject health information is not acceptable.

17.   In whole manuscript there is no any test for sensitivity-that is not proper title of the manuscript-“ new sensitive biomarker”

Author Response

Reviewer #3

Comments 1: Part 2.1- what for was this part – it is known that obese animals have higher weight and higher organ weight, what is new in this part?

Answer: Thank you for your valuable comments. In this part, it was reconfirmed that body weight and kidney weight increased in obese animals as indicated previous studies.

Comments 2:   Lines 97-99- this sentence is not proper in English and difficult to understand; if there is no any damages- there is no morphological changes, there are two time ‘rats’ in one sentence.

Answer: Thank you for your valuable comments. We revised this parts as your comments. “However, glomerular and proximal tubular injuries, such as glomerular hyaline droplets, increased mesangial matrix, basophilic renal tubules, hyaline casts, and tubular dilatation, increased in HFD-fed ZDF rats”

Comments 3:  Line 102- the last part of the sentence is not logical- I think You mean ‘…identified in HFD-fed ZDF rats’ ?

Answer: Yes, thank you for your valuable comments. We revised discussion parts as your comments.

Comments 4: Figure 2 part B- it isn’t understandable. Why on the figure are three parts, what is mean?

Answer: Thank you for your valuable comments. In Figure 2B we indicated the specific damage in the renal tissues after HFD-fed rats “Histology scores for renal injury are as follows: 1, normal; 2, minimal; 3, slight; 4, moderate; 5, marked; and 6, severe at 5 weeks, 12 weeks, and 19 weeks in the kidney of HFD-induced rats with DN”.

Comments 5:  Table 1- this groups are not comparable, control is too small for statistic, the is lack of clinical characteristic, what is not proper for conclusions. There is mistake “BUM”, as I think? How was the DN diagnosed in this group, and how exclude other reasons of kidney damages.

Answer: Thank you for your valuable comments. As your comments, the control subjects number are very small because we could not recruiting a large number of subject from the old samples. The BUM is mistake and we corrected BUN. Thank you for your kind comments. That’s right. In this study, this is a preliminary research result proving that the PKM2 protein detected in the urine can be measured earlier than the existing renal toxicity biomarkers such as BUN and SCr. Thus, this study proves that PKM2 is detected in urine when BUN and SCr are not elevated in DM patients.

Comments 6:   Line 143- according to inclusion criteria- where they are written? If in control group are almost only men, it is normal that will be higher mean BUN

Answer: Thank you for your valuable comments. In patients with diabetes, the mean age and duration of diabetes were 65.8 and 3 years, respectively, with BUN levels that were significantly more than normal values. Other clinical characteristics of the patients with diabetes were not detected, since patient information was obtained from an old sample. We did not calculate separately the BUN levels between women and man.

Comments 7:   Lines 145-146- it couldn’t be compared if we don’t have any information about subjects enrolled to control group.

Answer: In patients with diabetes, the mean age and duration of diabetes were 65.8 and 3 years, respectively, with BUN levels that were significantly more than normal values. Other clinical characteristics of the patients with diabetes were not detected, since patient information was obtained from an old sample.

Comments 8:    Lines 148-150- Did You exclude from the study patients with different other kidney diseases, what are the exclusion criteria for the study. If we want to find the early biomarker, we should have series of tests in the same group of patients and then compare what were different in group which later develop DN. All mention markers are controversial, and are not approved, despite for example NGAL is known for many years.

Answer: Thank you for your valuable comments. Yes, we exclude the study patients with different other kidney diseases, only we included diabetic patients. As your comments, test with the same group of patients is very important to develop DN biomarkers in clinical samples. In the future, large-scale in-depth examinations of the viability of urinary PKM2 protein as a prognostic biomarker of DN are warranted to confirm the hypothesis of this study.

Comments 9:   Figure 4- If Authors compare to BUN- that marker seems not be very early but rather late.

Answer: Thank you for your valuable comments. That’s right, but in HFD-fed ZDF rats model, we find that urinary excretion of PKM2 levels were clearly observed after 5 weeks HFD-fed rats without changes in BUN and SCr levels.

Comments 10:  Lines 200—203- I couldn’t find information that Authors took the samples at 9 weeks.

Answer: Thank you for your valuable comments. We revised discussion parts “As we have demonstrated, although urinary albumin levels were not increased in HFD-induced diabetic rats after 5 weeks, the levels significantly increased after 12 weeks.”

Comments 11:   Lines 205-206- I couldn’t find information about taking samples at 9 weeks, and their results.

Answer: Thank you for your valuable comments. We revised discussion parts as your comments. ]. Our data revealed that these renal injury biomarkers (BUN and SCr) are not increased in the serum of HFD-fed rats collected 5 weeks before a diagnosis of DN. However, PKM2 levels were significantly increased in the urine of rats 5 weeks after HFD administration.

Comments 12:   Lines 224-226- there is no bibliography to this sentence

Answer: Thank you for your valuable comments. We added the reference.

Comments 13   Part 4.1- there is lack of diabetes recognition criterion in rats

Answer: Thank you for your valuable comments. In this experiment, Blood glucose concentration was measured every week, and when it was over 200, it was evaluated as inducing diabetes.

Comments 14:  Lines 238-239- There is no information how the blood samples were taken

Answer: Thank you for your valuable comments. We added in 4.3 “Blood was collected from the tail vein in the morning, and blood glucose levels were estimated using a glucometer (ACCU-CHEK; Daeil Pharm. Co, Ltd., Seoul, Korea)

Comments 15:   Lines 239-241- there is no information at what time point were urine samples collected

Answer: Thank you for your valuable comments. Urine was collected from each animal kept in the metabolic cage overnight, and 24 h urine samples were collected at 5, 12, and 19 weeks from individual animals in containers (0.1% sodium azide).

Comments 16:   Line 300- How many control subjects You enroll to the study (in the text are 20, in the table 8 person. What are inclusion and exclusion criteria. These are not comparable groups, and lack of subject health information is not acceptable.

Answer: Thank you for your valuable comments. We corrected with 8 normal subject. We do not available health information from normal subject.

Comments 17:   In whole manuscript there is no any test for sensitivity-that is not proper title of the manuscript-“ new sensitive biomarker

Answer: Thank you for your valuable comments. We corrected this part “not sensitive” early biomarker as your comment.

Reviewer 4 Report

Dr. Park and colleagues presented a promising research of diabetic kidney disease. The main strength is that this study contains both animal and clinical part with advanced techniques. However, the evidence presented into the current draft is not sufficient to be published in IJMS. The article also contains significant inaccuracies in the text, some of them cannot consider as acceptable. Major revisions could improve the manuscript according to the following items.

1. Relating to the design of the experiment

1.1. The authors set up an aim of the research to provide an evidence that PKM2 is a marker for early diabetic kidney disease. The authors mentioned early detection as detection increased PKM2 in patients/animals with stage of diabetic kidney diseases without elevated serum creatinine or BUN (Line 224–225). They also mentioned the problem of increased prevalence of decreasing renal function before development of albuminuria in patients with diabetes (Line 40–42). Nevertheless, according to the results of the experiment, HFD rats were characterized elevated BUN, serum creatinine and albuminuria. So, the conclusion that PKM2 may be considered as an early marker of diabetic kidney disease was not supported by the current results of the experiment.

1.2. The authors also aimed to prove that PKM2 is a sensitive marker of diabetic kidney disease. While the current study did not estimate of sensitivity or specificity of PKM2 for diagnosis of diabetic kidney disease.

1.3. Some of examinations in the research plan looks excessive because don’t follow the research goal. For example, providing the results of pancreas and AGE concentrations without link to diabetic kidney disease can be omitted. The measurements of testicle weight, serum ALT and AST, assessing AGE with VetScan looks to be not necessarily according to the research goal as well.

1.4. The control group does not match to the diabetic patients for age and sex in the clinical part of the research. According to the results is not clear if the found changes resulted from diabetes or can be associated with age or female sex.

1.5. The authors mentioned, that PKM2 has been previously studied (Line 206–214). According these, what is the novelty of the current research?

2. Relating to animal research

2.1. The results of the study did not match the description in the Materials and Methods section. E.g., the authors presented the results of WB for urinary biomarkers concentrations (Line 117–132), while they provided the protein isolation protocol from renal cortex (Line 287–288). The authors also measured testicles weight (Line 247), serum ALT and AST activities (Line 258), AGE levels with the VetScan analyzer (Line 260), PAS-stained sections of renal cortex (Line 281), but these results are not presented in the manuscript.

2.2. According to the Materials and Methods as well as the Result sections is not clear, which week urine was used to assess the urinary albumin and studied biomarkers excretions. The providing of 24h albumin excretion or albumin-to-creatinine ratio instead of urinary albumin concentrations could be more following to current methods of assessment of albuminuria.

2.3. The methods for semiquantitative estimation for changes in renal tissues (Fig. 2B) are not described.

2.4. In the discussion section, authors mentioned “As shown in Figure 2, the urinary excretion of PKM2 weeks after HFD was insignificant, whereas that at 9–15 weeks significantly increased, suggesting that renal PKM2 started to leak into urine at 9 weeks and continued to leak until at  least 15 weeks” (Line 201-204). However, Fig. 2 does not contain the results related to urinary excretion of PKM2. Fig. 3 also contain IHC images.

2.5. The protocols used for IHC, as well as used antibodies, are not presented in the article. Semoquantitative or quantitative estimation of PKM2 and KIM-1 areas or intensity of staining can provide an additional evidence to the results. Fig. 3, PKM2 and KIM-1 in ND rats: please reassure if the 50 μm line bar is exact (these picture looks with lower magnification comparing with pictures for HFD rats). The pictures with secondary antibody only control (no primary antibody control, negative control) must be provided.

2.6. WB protocol must be added with information about antibody used into the research and the protocol protein isolation from urine samples. Adjustment of found results to the urinary creatinine concentrations could be more following to current methods of assessment for urinary protein excretion.

2.7. The authors mentioned that number of animals in each ND and HFD groups was six. While, Fig. 3 contains renal images for 5, 12 and 19 Week. Consequently, the number animals used for IHC for each weak were two. Body weight and blood glucose for animal after Week 5 were also measured with fewer number of animals (N=4 from Week 5, and N=2 from Week 12). Using ARRIVE 2.0 guideline to present the animal part of the research can provide cleared description. The evidence if these number of animals is sufficient for statistical significance must be provided. Statistical section must be added by the preliminary estimation of number animals in each group.

2.8. The authors used parametrical tests, while they did not mention if the data distribution was normal or not. The statistical section must be added with tests for normality of distribution of non-parametrical test for non-normally distributed data.

3. Relating to clinical part of the research

3.1. The description of these part does not contain some important information, such as design, inclusion and exclusion criteria. Using the STROBE guideline may provide a better evidence for found results. Table 1 does not contain serum creatinene levels. The authors claimed that “Other clinical characteristics of the patients with diabetes were not detected owing to the information of patients available from an old sample” (Line 145–146). However, calculated eGFR and measurement of UACR in the urine sample used for WB can provide a crucial information to the study. Another important information related to the research is the duration of diabetes, diagnosis (type 1 or type 2 diabetes), arterial hypertension, smoking, race and other renal disease can be significant as well.

3.2. The presentations of results for urinary excretion of PKM2 and other markers for each patient is quite difficult to understand. The providing medians with interquartile range or mean with SD could be a better option. Following statistical procedures are critical as well. The associations of PKM2 with serum creatinine, eGFR and albuminuria can provide a better evidence to the results.

3.3. Similar to 2.6 for Animal research.

4. Inaccuracies and other remarks

4.1. The mismatches between the Results and the Materials and Methods sections make the manuscript difficult to understand. The manuscript also have a lot of sentences in blue, while according to the IJMS standards the text generally must be typed in black. Line 48 contain a link to some reference, but this reference was not mentioned. Line 167–168 also contain a disambiguous typo (“early detection of DN can be delayed by developing ESRD”).

4.2. Some reference were cited inaccurately. E.g. the authors mentioned that “DN occurs in approximately 30–40% of patients with type 2 diabetes and 15–25% patients with type 1 diabetes” according to the Reference 26. However, this reference provided other frequencies: 40% and 30%, respectively. The authors also referred KDIGO 2012 criteria for diagnosis of diabetic nephropathy in rats (Line 206, Reference 46), while this guideline relates to chronic kidney disease in humans.

Author Response

1: Relating to the design of the experiment

Comments 1.1:  The authors set up an aim of the research to provide an evidence that PKM2 is a marker for early diabetic kidney disease. The authors mentioned early detection as detection increased PKM2 in patients/animals with stage of diabetic kidney diseases without elevated serum creatinine or BUN (Line 224–225). They also mentioned the problem of increased prevalence of decreasing renal function before development of albuminuria in patients with diabetes (Line 40–42). Nevertheless, according to the results of the experiment, HFD rats were characterized elevated BUN, serum creatinine and albuminuria. So, the conclusion that PKM2 may be considered as an early marker of diabetic kidney disease was not supported by the current results of the experiment.

Answer: Thank you for your critical comments. As you mentioned, we did not indicated PKM2 levels is detected early in the serum BUN and sCr levels in this study. Therefore, we corrected this part “For instance, in rats with streptozotocin-induced diabetes, we found that the urinary excretion of PKM2 was significantly increased, suggesting that PKM2 plays a detrimental role in DN.”

Comments 1.2. The authors also aimed to prove that PKM2 is a sensitive marker of diabetic kidney disease. While the current study did not estimate of sensitivity or specificity of PKM2 for diagnosis of diabetic kidney disease.

Answer: Thank you for your kind comments. We added a few data to comparing early or sensitive biomarkers for DN.

Comments 1.3. Some of examinations in the research plan looks excessive because don’t follow the research goal. For example, providing the results of pancreas and AGE concentrations without link to diabetic kidney disease can be omitted. The measurements of testicle weight, serum ALT and AST, assessing AGE with VetScan looks to be not necessarily according to the research goal as well.

Answer: Thank you for your valuable comments. We delete serum ALT and AST

Comments 1.4. The control group does not match to the diabetic patients for age and sex in the clinical part of the research. According to the results is not clear if the found changes resulted from diabetes or can be associated with age or female sex.

Answer: In this time, we could be used only small size of samples for clinical trials. Next, ~~

Comments 1.5. The authors mentioned, that PKM2 has been previously studied (Line 206–214). According these, what is the novelty of the current research?

Answer: It has been proven that the PKM2 content released from urine in the presence of kidney lesions is very good as an early diagnostic marker for kidney disease. However, the research team identified for the first time whether PKM2 can be detected in clinical samples of diabetic patients.

  1. Relating to animal research

Comments 2.1. The results of the study did not match the description in the Materials and Methods section. E.g., the authors presented the results of WB for urinary biomarkers concentrations (Line 117–132), while they provided the protein isolation protocol from renal cortex (Line 287–288). The authors also measured testicles weight (Line 247), serum ALT and AST activities (Line 258), AGE levels with the VetScan analyzer (Line 260), PAS-stained sections of renal cortex (Line 281), but these results are not presented in the manuscript.

Answer: Thank you for your excellent comments. This part is our mistake we carefully changed this partThe levels of various urinary protein-based markers, such as KIM-1 (1:1000), NGAL (1:1000), and PKM2 (1:1000) were analyzed using Western blotting. Briefly, the urine was centrifuged (1000×g for 10 min), and the supernatant (considered to contain the total urinary proteins) was collected and diluted in a 1:2 ratio with double distilled water (DDW).”

Comments 2.2. According to the Materials and Methods as well as the Result sections is not clear, which week urine was used to assess the urinary albumin and studied biomarkers excretions. The providing of 24h albumin excretion or albumin-to-creatinine ratio instead of urinary albumin concentrations could be more following to current methods of assessment of albuminuria.

Answer: Thank you for your excellent comments. In this study, we measured the micrialbumin in the 24 h collected urine at 19 weeks. These parameters only confirm the HFD-induced diabetic nephropathy.

Comments 2.3. The methods for semiquantitative estimation for changes in renal tissues (Fig. 2B) are not described.

Answer: We described Fig. 2B in the results part.

Comments 2.4. In the discussion section, authors mentioned “As shown in Figure 2, the urinary excretion of PKM2 weeks after HFD was insignificant, whereas that at 9–15 weeks significantly increased, suggesting that renal PKM2 started to leak into urine at 9 weeks and continued to leak until at  least 15 weeks” (Line 201-204). However, Fig. 2 does not contain the results related to urinary excretion of PKM2. Fig. 3 also contain IHC images.

Answer: We are really sorry that we carefully checked and changed this part as your valuable comments.

Comments 2.5. The protocols used for IHC, as well as used antibodies, are not presented in the article. Semoquantitative or quantitative estimation of PKM2 and KIM-1 areas or intensity of staining can provide an additional evidence to the results. Fig. 3, PKM2 and KIM-1 in ND rats: please reassure if the 50 μm line bar is exact (these picture looks with lower magnification comparing with pictures for HFD rats). The pictures with secondary antibody only control (no primary antibody control, negative control) must be provided.

Answer: We added this part in Materials and method “

Comments 2.6. WB protocol must be added with information about antibody used into the research and the protocol protein isolation from urine samples. Adjustment of found results to the urinary creatinine concentrations could be more following to current methods of assessment for urinary protein excretion.

Answer: We added the protocol on the protein isolation in the urine.

Comments 2.7. The authors mentioned that number of animals in each ND and HFD groups was six. While, Fig. 3 contains renal images for 5, 12 and 19 Week. Consequently, the number animals used for IHC for each weak were two. Body weight and blood glucose for animal after Week 5 were also measured with fewer number of animals (N=4 from Week 5, and N=2 from Week 12). Using ARRIVE 2.0 guideline to present the animal part of the research can provide cleared description. The evidence if these number of animals is sufficient for statistical significance must be provided. Statistical section must be added by the preliminary estimation of number animals in each group.

Answer: Thank you for your critical comments. This is our mistake that we used all the same animal numbers each week (5, 12, 19 weeks., Thus, we used 18 animals in the HFD-induced rats models.

Comments 2.8. The authors used parametrical tests, while they did not mention if the data distribution was normal or not. The statistical section must be added with tests for normality of distribution of non-parametrical test for non-normally distributed data.

Answer: Thank you for your critical comments. We revised the statistical analysis on the all data.

Comments 3.1. The description of these part does not contain some important information, such as design, inclusion and exclusion criteria. Using the STROBE guideline may provide a better evidence for found results. Table 1 does not contain serum creatinene levels. The authors claimed that “Other clinical characteristics of the patients with diabetes were not detected owing to the information of patients available from an old sample” (Line 145–146). However, calculated eGFR and measurement of UACR in the urine sample used for WB can provide a crucial information to the study. Another important information related to the research is the duration of diabetes, diagnosis (type 1 or type 2 diabetes), arterial hypertension, smoking, race and other renal disease can be significant as well.

Answer: Thank you for your critical comments. As you pointed out, GFR and urinary creatinine analysis can provide very important information in the analysis of clinical samples of diabetic patients. The clinical sample used in this study was provided as part of a preliminary study because a newly designed sample for this study could not be used. Therefore, it was not possible to know the exact information about various clinical samples, but we wanted to see if PKM2 protein could be detected in the urine of diabetic patients. These findings are the first to provide evidence in human urine samples that urinary excretion of PKM2 is a feature of DN. Future analyses will test whether PKM2 excretion is closely associated with DN progression and urinary metabolic biomarkers in patients during the study. It waived the need for informed consent because of the retrospective nature of the study and the lack of personal information.

Comments 3.2. The presentations of results for urinary excretion of PKM2 and other markers for each patient is quite difficult to understand. The providing medians with interquartile range or mean with SD could be a better option. Following statistical procedures are critical as well. The associations of PKM2 with serum creatinine, eGFR and albuminuria can provide a better evidence to the results.

Answer: Thank you for your kind comments. That’s right. In this study, this is a preliminary research result proving that the PKM2 protein detected in the urine can be measured earlier than the existing renal toxicity biomarkers such as BUN and SCr. Thus, this study proves that PKM2 is detected in urine when BUN and SCr are not elevated in DM patients.

Comments 4.1. The mismatches between the Results and the Materials and Methods sections make the manuscript difficult to understand. The manuscript also have a lot of sentences in blue, while according to the IJMS standards the text generally must be typed in black. Line 48 contain a link to some reference, but this reference was not mentioned. Line 167–168 also contain a disambiguous typo (“early detection of DN can be delayed by developing ESRD”).

 Answer: We corrected these parts and revised as your comments.

Comments 4.2. Some references were cited inaccurately. E.g. the authors mentioned that “DN occurs in approximately 30–40% of patients with type 2 diabetes and 15–25% patients with type 1 diabetes” according to the Reference 26. However, this reference provided other frequencies: 40% and 30%, respectively. The authors also referred KDIGO 2012 criteria for diagnosis of diabetic nephropathy in rats (Line 206, Reference 46), while this guideline relates to chronic kidney disease in humans.

Answer: Thank you for your valuable comments. We carefully checked and corrected all references in this paper. We delete this part “Although not all patients with diabetes progress to DN, it has been estimated that DN affects 30–50% of diabetic patients.” The reference 46 changed with “Ref. 44. Deji N., Kume S., Araki S., et al. Structural and functional changes in the kidneys of high-fat diet-induced obese mice. Am. J. Physiol. Renal Physiol. 2009, 296(1), F118–F126.”`

Round 2

Reviewer 3 Report

Dear Authors, 

In the manuscript I still didn't find the proper inclusion and exclusion criteria. This is really not understandible that You present study with so small control group, study will be much more worth if You add even 7-12 person to control group, matched in age, sex etc. 

Author Response

Reviewer #3

Comment: In the manuscript I still didn't find the proper inclusion and exclusion criteria. This is really not understandible that You present study with so small control group, study will be much more worth if You add even 7-12 person to control group, matched in age, sex etc. 

Answer: Thank you for your critical comments on the number of normal subjects in this study. As your comments, too a few normal subjects (20 members) groups in this study are not sufficient to understand the data reliable. We totally agree with your comments. However, in this study, we try to find out to only evaluate whether PKM2 can be detected in the urine of diabetic patients. As indicated in Figure 4A, PKM2 levels were not detected at all in the urine of normal subject as well as in the urine of normal diet animals but not diabetic rats. Therefore, we did not including an additional large number of normal subjects in the present study. However, selecting only young subjects as normal control to have failed to make the essence of this study transparent. Therefore, it is expected that further study considering with the age-adjusted normal and diabetic patients groups must be performed in the future.

Reviewer 4 Report

The authors changed the manuscript according to the Reviewers' notes. However, further minor remarks can be suggested to change:

1. The results of urinary creatinine presented on Fig. 3A looks quite unusual and unexpected. For animals and people with diabetes is more common to have less concentrated urine with lower concentrations of creatinine as compared with animals and people without diabetes. This parameter generally depends upon water consuption. Please, double check this graph.

Moreover, it is not nessesserly to present the urine creatinene concentrations. While, it is more intresting to present urinary albumin-to-creatinine and protein-to-creatinine ratios. These are wildly used in clinical pratice and provide a better discribtion of the model.

2. The draft contains no information about supplier and catlogue numbers of antibodies used in the research. Please add these information to the draft as it is crutual for tranparancy and make the results more trustworthy.

After these minor changes, the muscript can be published without additional review.

Author Response

Reviewer #4

The authors changed the manuscript according to the Reviewers' notes. However, further minor remarks can be suggested to change:

Comment 1. The results of urinary creatinine presented on Fig. 3A looks quite unusual and unexpected. For animals and people with diabetes is more common to have less concentrated urine with lower concentrations of creatinine as compared with animals and people without diabetes. This parameter generally depends upon water consuption. Please, double check this graph. Moreover, it is not nessesserly to present the urine creatinene concentrations. While, it is more intresting to present urinary albumin-to-creatinine and protein-to-creatinine ratios. These are wildly used in clinical pratice and provide a better discribtion of the model.

Answer: Thank you for your critical comments. As your comments, we carefully checked urinary biochemical parameters. We corrected the urinary creatinine levels. “We observed that urinary albumin to creatinine ratio was significantly higher in HFD-induced diabetic rats than in ND-fed rats (Figure 3A). Protein levels in the urine significantly increased with progressing hyperglycemic conditions, accompanied by significant decreases in urinary Cr, which indicated a significant alternation in the kidney function of HFD model rats (Figure 3A).”

Comment 2. The draft contains no information about supplier and catlogue numbers of antibodies used in the research. Please add these information to the draft as it is crutual for tranparancy and make the results more trustworthy. After these minor changes, the muscript can be published without additional review.

Answer: As your comments, we added the catlogue numbers of antibodies used in the research. Antibody against KIM-1 (PA5-79345) was purchased from Invitrogen (Waltham, MA, USA). NGAL (ab23477) and PKM2 (ab85555) antibodies was purchased from Abcam (Cambridge, Massachusetts, USA), while anti-mouse and anti-rabbit IgG, HRP-linked-conjugated secondary antibodies were procured from Santa Cruz Biotech. (Santa Cruz, California, USA).